# Association of Vasopressors Dose Trajectories with Enteral Nutrition Tolerance in Patients with Shock: A Prospective Observational Study

**DOI:** 10.3390/nu14245393

**Published:** 2022-12-19

**Authors:** Luping Wang, Tao Zhang, Hua Yao, Qian Xu, Xin Fu, Jing Yang, Bo Wang, Zhongwei Zhang, Xiaodong Jin, Yan Kang, Qin Wu

**Affiliations:** 1Department of Critical Care Medicine, West China Hospital, Sichuan University, No. 37 Guo Xue Road, Chengdu 610041, China; 2Administrative Office, West China Tianfu Hospital, Sichuan University, Chengdu 610213, China

**Keywords:** NEQ, trajectory, shock, enteral nutrition, feeding intolerance

## Abstract

(1) Background: Studies on the long-term patterns of using vasopressors in patients with shock and their correlations with the risk of feeding intolerance (FI) are limited. This study aimed to characterize the norepinephrine equivalent dose (NEQ) trajectories and explore its correlations with FI in patients with shock. (2) Methods: This study prospectively enrolled patients with shock, who received vasopressors from August 2020 to June 2022. The Growth Mixed Model (GMM) was used to traverse longitudinal NEQ data at six-hour intervals and identify the latent trajectories of NEQ use in these patients. Cox proportional hazards regression models were used to examine the correlations of NEQ trajectories with FI. (3) Results: This study included a total of 210 patients with shock recruited from August 2020 to June 2022. Four trajectories of NEQ dose were identified and characterized by low-dose stable NEQ (L-NEQ, *n* = 98), moderate-dose stable NEQ (M-NEQ, *n* = 74), high-dose stable NEQ (H-NEQ, *n* = 21), and rapidly rising NEQ (R-NEQ, *n* = 17), with NEQ doses of 0.2, 0.4, 0.4, and 0.5 µg/kg/min at enteral nutrition (EN) initiation, respectively. The incidences of FI were 37.76%, 67.57%, 80.95%, and 76.47% in the L-NEQ, M-NEQ, H-NEQ, and R-NEQ groups, respectively (*p* < 0.001). As compared to the L-NEQ group, the risk of FI occurrence increased in the M-NEQ, H-NEQ, and R-NEQ groups (all *p* < 0.05). (4) Conclusions: The risk of FI was significantly associated with NEQ trajectories. It might be appropriate to initiate EN when the NEQ dose is stabilized below 0.2 µg/kg/min in patients with shock.

## 1. Introduction

Enteral nutrition (EN) support is an essential component of supportive care for critically ill patients [1,2]. Early EN has multiple benefits, such as the direct delivery of nutrients for nourishing the intestinal tract, maintaining the integrity of the intestinal barrier and function, and relieving oxidative stress and inflammatory response [1,3,4]. However, the unstable hemodynamic conditions and use of vasopressors might impair and exacerbate gastrointestinal hypoperfusion in patients with shock [1,4]. Inadequate EN delivery might further worsen this condition, causing severe gastrointestinal complications [5].

Numerous studies have demonstrated the feasibility and safety of EN in patients with shock; however, in clinical practices, the management of EN in these patients is still a difficult task [6,7,8,9,10,11,12,13,14,15,16]. Current guidelines offer vague and inconsistent recommendations [17,18,19,20,21,22]. In patients with shock, EN should be postponed until the hemodynamic condition is fully resuscitated and/or becomes stable as per the recommended guidelines [17,18]. Other guidelines and clinical experts suggest that the low-dose EN is recommended within 48 h after the intensive care unit (ICU) admission of patients with shock, receiving a small or moderate dose of vasopressors, or the consideration of EN only or maintenance EN at vasopressor dose equivalent scores > 12 [21,22]. However, ambiguous definitions and unvalidated scores might be dangerous for guiding clinical practices.

Vasopressors have various effects on the gastrointestinal tract [23,24,25]. Current literature indicates conflicting results regarding the feasibility and safety of EN in patients with shock. Previous studies showed that the safety cut-off value of EN was likely to be less than 0.3 µg/kg/min in these patients [6,7,8,9,10,11,12,13,14,16]. NUTRIREA-2 study, a large multicenter randomized controlled trial, reported that early EN with 0.56 µg/kg/min norepinephrine infusion was associated with an increased risk of gastrointestinal adverse events in the patients with shock [26]. Interestingly, the NUTRIVAD study, a recent multicenter observational study, reported that the EN with a norepinephrine dose > 0.5 µg/kg/min was feasible and safe in the patients [15]. These differences might be due to the differences in the time points of norepinephrine equivalent dose (NEQ) extraction and selection of outcome indicators as well as the unknown time effects of NEQ dose on the intestinal injury. Due to the unpredictable effects of vasopressors on intestinal circulation, the tolerance of EN might change in different subgroups with the temporal effects of NEQ dose. However, studies describing the correlations between the long-term patterns of NEQ dose and EN tolerance in patients with shock are limited. Understanding the NEQ trajectories might provide useful guidance for clinical practice and nursing care. Therefore, a prospective study, applying Growth Mixed Model (GMM), was conducted to characterize the NEQ trajectories and explore its correlations with feeding intolerance (FI) in patients with shock.

## 2. Materials and Methods

### 2.1. Study Design

This prospective study was performed in a 50-bed central intensive care unit (ICU) of the West China Hospital located in Chengdu, Sichuan Province, China, from August 2020 to June 2022. The study protocol was reviewed and approved by the West China Hospital of Sichuan University Biomedical Research Ethics Committee (No. 2021S673). All the enrolled patients or their next of kin provided written informed consent.

### 2.2. Patients

The patients with shock, requiring vasopressors and receiving EN support, were recruited. The shock was defined as a life-threatening systemic acute circulatory failure, associated with the inadequate utilization of oxygen by cells (blood lactate levels exceeding 2 mmol/L) accompanied by a series of signs of inadequate tissue and organ perfusion [27]. The criteria for the exclusion of patients were as follows: (1) the patients who had stopped vasopressors before starting EN support; (2) the patients who had undergone gastrointestinal surgery within a month; (3) the patients who had active gastrointestinal bleeding; (4) the patients who died within in 24 h after EN initiation; (5) the patients who already had EN support before recruitment; (6) the patients who had severe acute pancreatitis; (7) the patients who refused to participate in this study; (8) the patients who had contraindications to EN, such as intestinal obstruction or ischemia; (9) the patients who were less than 18 years old; and (10) the patients who were pregnant.

Upon ICU admission, the basic characteristics of the patients, including their diagnoses, types of shock, Acute Physiology and Chronic Health Evaluation II (APACHE II) score, and Nutritional Risk Screening (NRS) 2002 score, were recorded and assessed. Laboratory data were collected at the time of EN initiation. Vasopressors (norepinephrine, adrenaline, dopamine, and vasopressin) were recorded every hour after EN initiation, and NEQ was calculated according to Equation (1) [28].
NEQ = [norepinephrine (μg/kg/min)] + [dopamine (μg/kg/min) ÷ 150] + [adrenaline (μg/kg/min)] + [phenylephrine (μg/kg/min) ÷ 10] + [vasopressin (unit/min) ÷ 0.4](1)

The information about therapeutics, including the average daily dose of sedation and analgesia drugs, inotropic therapy, prokinetics, continuous renal replacement therapy, and extracorporeal membrane oxygenation, was collected throughout the follow-up. In addition, the cumulative fluid balance was collected during the first week in the ICU.

### 2.3. Nutrition Strategy

The PEP uP (Enhanced Protein-Energy Provision via the Enteral Route in Critically Ill Patients) protocol was applied as the main EN strategy [29]. The enteral formulas were polymeric (0.9 kcal/mL) and semi-elemental (0.8 kcal/mL) formula in the whole cohort. On the first day, the initial feeding rate was set to 10 mL/h. The patient’s feeding tolerance was assessed daily. If EN was well tolerated, the feeding rate was adjusted by increasing the feeding rate according to the patient’s condition, but did not exceed 150 mL/h. On the second or subsequent days, if applicable, the 24-h volume goal was calculated by clinicians based on the PEP uP protocol. Parenteral nutrition supplementation and the use of prokinetics drugs were not routine options; clinicians could choose to use them based on the patient’s gastrointestinal motility, EN tolerance, and nutritional status. The protein target was 1.2–2.0 g/kg/day.

### 2.4. Outcomes Measurement

FI was defined as the interruption of EN due to the presence of one of the following indications: vomiting/regurgitation, diarrhea, ileus, and suspected mesenteric ischemia/perforation. Any visible reflux of gastric contents was diagnosed as vomiting/regurgitation. Diarrhea was defined as loose or liquid stools ≥ 3 times per day with stool volume > 250 mL/day. The diameter of the cecum > 9 cm or colon > 6 cm was diagnosed as ileus. In case of sudden abdominal pain, abdominal distention, peritonitis/muscular defense, gastrointestinal bleeding, multiple organ dysfunctions/multiple organ failure, and positive radiographic signs, such as expanded and thickened loops of the intestine with thumbprinting, the air in the intestinal wall, portal gas, and air in the peritoneal cavity, the condition was defined as suspected mesenteric ischemia [3,30]. Abdominal computed tomography was used to confirm mesenteric ischemia [11,15]. After EN initiation, the EN tolerance and FI details in the patients were assessed and recorded daily. FI was independently diagnosed by three physicians with extensive clinical experience.

The primary endpoint was the occurrence of FI. The secondary endpoints were the incidences of FI in the first 3 days, 3 to 7 days, 7 to 14 days, and >14 days after EN initiation, 28-day all-cause mortality, EN intake, total nutrition intake, hospital-acquired infections, mechanical ventilation (MV) duration, and duration of ICU stay.

### 2.5. Statistical Analyses

The latent trajectories of NEQ were defined in the first 7 days after EN initiation using GMM. The GMM, also named latent class trajectory modeling, can traverse the longitudinal data, and divide heterogeneous populations into more homogeneous clusters or categories while considering the random effects of individuals within the same trajectory group [31]. The longitudinal NEQ data at six-hour intervals after EN initiation were set as a linear (linear term of time) or nonlinear function of time (quadratic or cubic term of time), and the GMM model was constructed by fitting the time function from the linear term to the cubic term of time and fitting the model with trajectories in groups 1 to 6 for each functional form, respectively. The best fitting model was identified as the one with lower Bayesian information criterion (BIC) and Akaike information criterion (AIC), entropy close to zero, and relative entropy close to 1. Meanwhile, the mean posterior probabilities were greater than 0.7, and the proportion of patients with posterior probabilities >70% was greater than 65% in each group. The missing values were inserted using the linear interpolation method. GMM was performed using the R package “lcmm” (version 1.9.5) and “LCTMtools” (version 0.1.3) [31,32].

Descriptive statistics were expressed as mean (standard deviation, SD) or median (interquartile range, IQR) for the continuous variables and as numbers (percentages) for categorical variables. Student t-test and Mann–Whitney U-test were used to test the differences in continuous variables, and the chi-squared test and Fisher’s exact test were used for the differences in categorical variables. The differences between the patients with and without FI in each trajectory were compared to explore the potential heterogeneity factors in each trajectory. COX proportional hazard model was used to explore the correlations of NEQ trajectories with the occurrence of FI and mortality of shock patients while correcting for the covariates, such as age, body mass index, NEQ base dose at EN initiation, APACHE II score, lactic acid contents, dobutamine dose during the whole disease progress, and cumulative fluid balance. These correlations were evaluated in the unadjusted and adjusted models. In each model, the median hazard ratio (HR) was used for each group as a continuous variable in a linear regression model, and the linear trend of each group in increasing risk was tested. Kaplan–Meier survival curve analysis was performed to identify the cumulative risk for FI occurrence and 28-day mortality, and the log-rank test was used to test the differences among the groups. All the statistical analyses were performed using the R package 4.0.3 (https://www.r-project.org/, accessed on 14 November 2021), RStudio (https://www.rstudio.com/, accessed on 14 November 2021), and IBM SPSS 26.0 (IBM Corp., Armonk, NY, USA). The simplified workflow of the study was shown in Appendix A.

## 3. Results

### 3.1. Participants

A total of 2719 patients were admitted and screened in our center from August 2020 to June 2022. After applying the exclusion criteria, a total of 210 eligible patients with shock were included in this study (Figure 1). The median (IQR) age of all the included patients was 64 (49.8–72). The male patients accounted for 67.14% (*n* = 141) of all the included patients with a median (IQR) BMI of 22.9 (20.2–24.8). Moreover, the median (IQR) APACHE II score of all the patients was 20.9 (14–26), and 178 (84.76%) patients had sepsis shock (Table 1). 

### 3.2. Baseline Characteristics of Patients in Each Trajectory Group

The model fitting results are provided in Appendix A. Consistent with the results of GMM, the model with four trajectory classes was the best by showing lower BIC (1313.30), AIC (1249.70), and entropy (43.36), and relative entropy close to 1 (0.85). The trajectories of NEQ use in patients with shock are shown in Figure 2. Class 1 (*n* = 98), Class 2 (*n* = 74), and Class 3 (*n* = 21), which were named “low-dose stable NEQ” (L-NEQ), “moderate-dose stable NEQ” (M-NEQ), and “high-dose stable NEQ” (H-NEQ), respectively, showed stable NEQ at low, medium, and high levels in the patients with shock within 7 days after EN initiation. Class 4 (*n* = 17), which was named “rapidly rising NEQ” (R-NEQ), showed a rapidly increasing NEQ use in the patients with shock after 7 days of EN initiation.

The R-NEQ group was considered a more severe form of the disease and worse circulation because most of these patients had higher lactic acid contents (2.4 mmol/L), received a higher dose of NEQ (0.5 μg/kg/min), sedation and analgesia drugs (midazolam: 3.8 mg/kg/day, sufentanyl: 2.6 μg/kg/day), and dobutamine (0.6 μg/kg/day), and a higher positive fluid balance in the first week (5.4 L). The H-NEQ group patients were characterized by more severe inflammatory responses due to higher levels of C-reactive protein (CRP) (162 mg/L), procalcitonin (10.3 μg/L), and interleukin-6 (140 μg/L). In contrast, the L-NEQ group patients were characterized by lower levels of CRP (93 mg/L), procalcitonin (1.6 μg/L), and interleukin-6 (64.8 μg/L), and were administered with lower NEQ dose at EN initiation (0.2 μg/kg/min). The M-NEQ group patients had inflammatory and circulatory profiles in between those of the H-NEQ and L-NEQ group patients (Table 1, Figure 3). The hourly changes in the NEQ in the patients with shock are provided in Appendix A. The differences in the characteristics between the patients with and without FI in each group were presented in Appendix A.

### 3.3. Clinical Outcomes

The R-NEQ group received significantly lower energy intake (5.5 kcal/kg/day vs. 8 kcal/kg/day vs. 9.9 kcal/kg/day vs. 7.7 kcal/kg/day, *p* = 0.017) and protein intake (0.2 g/kg/day vs. 0.3 g/kg/day vs. 0.4 g/kg/day vs. 0.3 g/kg/day, *p* = 0.014) as compared to the other three groups. The numbers (percentages) of patients with FI were 37 (37.76%) in the L-NEQ group, 50 (67.57%) in the M-NEQ group, 17 (80.95%) in the H-NEQ group, and 13 (76.47%) in the R-NEQ group (*p* < 0.001). The 28-day mortality was 15 (17.35%) in the L-NEQ group, 28 (37.83%) in the M-NEQ group, 13 (61.9%) in the H-NEQ group, and 17 (100%) in the R-NEQ group (Table 2). Moreover, the occurrences of FI within 3 days and 7–14 days after EN initiation were 11.22% and 9.18%, 23.3% and 18.92%, 23.81% and 28.57%, and 58.82% and 0%, in the L-NEQ, M-NEQ, H-NEQ, and R-NEQ groups, respectively (*p* < 0.001 and *p* = 0.019 for FI occurrence within 3 days and 7–14 days, respectively) (Figure 4). The differences in the clinical outcomes between the patients with and without FI in each group are summarized in Appendix A.

### 3.4. Cox Regression and Kaplan–Meier Survival Curve Analyses of Trajectories

The results of the Cox regression analysis of the patients with FI are listed in Table 3. In the model with unadjusted covariates, the M-NEQ, H-NEQ, and R-NEQ groups showed a higher risk of FI occurrence as compared to the L-NEQ group (HR (95% confidence interval [CI]): 2.226 (1.453, 3.409), 2.974 (1.67, 5.296), and 4.258 (2.252, 8.053), *p* < 0.001)). Models 2, 3, and 4 also showed similar results (*p* < 0.001 for all). For the mortality of patients, Cox regression analysis showed similar results (Appendix A). Kaplan–Meier analysis demonstrated that as compared to the other trajectory groups, the L-NEQ group had a higher probability of EN tolerance and a lower cumulative risk of FI (*p* < 0.001) (Figure 5) as well as higher survival probability and lower cumulative risk of mortality (*p* < 0.05) (Appendix A).

## 4. Discussion

In this study, GMM was used to explore the latent trajectories of NEQ use in patients with shock after EN initiation. Four trajectories of NEQ use were identified in these patients and were characterized as L-NEQ, M-NEQ, H-NEQ, and R-NEQ. As compared to the L-NEQ group, the occurrence of FI increased in the M-NEQ, H-NEQ, and R-NEQ groups in ascending order. The patients in the R-NEQ group had the highest incidence of FI in a short time. The H-NEQ group patients had a higher FI occurrence within 2 weeks as compared to that in the M-NEQ group patients, although there was no statistical difference. The risk of FI was significantly associated with the NEQ trajectories.

Not all critically ill patients obtain benefit from EN support. In patients with shock, intestinal hypoperfusion and vasopressors can increase the risk of intestinal mucosal ischemia [3,22]. Inadequate EN delivery might further aggravate the burden on the gastrointestinal tract [33]. EN delivery in patients with shock is complicated and obscure. An observational study reported that in the patients, receiving low-, moderate-, and high-dose noradrenaline, early EN with >0.3 µg/kg/min noradrenaline infusions did not have survival benefits [8]. The NUTRIREA-2 study indicated that the early EN support with 0.56 µg/kg/min norepinephrine infusion could increase the risk of gastrointestinal adverse events in patients with shock [26]. In contrast, the latest observational study, the NUTRIVAD study reported that EN support with a norepinephrine dose > 0.5 µg/kg/min was feasible and safe in the patients [15]. The NUTRIVAD study did not report the hemodynamic status during EN support in patients with high norepinephrine (>0.5 µg/kg/min); however, the norepinephrine dose decreased to 0.1–0.2 µg/kg/min after the first 7 days of ICU admission in all the patients. Numerous observational trials demonstrated that early EN support was feasible and safe in patients with shock. The studies also indicated that the safety threshold for NEQ was less than 0.3 μg/kg/min [6,7,8,9,10,11,12,13,14,15,16]. However, they only reported the dose of norepinephrine or NEQ at a single time point and did not describe the hemodynamic status during the entire EN support. Indeed, it is more meaningful to track the temporal changes in vasopressors dose during EN support than considering the impact of vasopressors dose at a single event point on the patient outcomes.

GMM is a data-driven approach, which identifies multiple unobserved subpopulations, depicts longitudinal variation within each subpopulation in easy-to-understand graphs, and examines the differences in variations among subpopulations [34]. This approach was used to identify four trajectories of NEQ in the patients with shock after EN initiation. The R-NEQ group included patients with rapidly progressing disease, requiring NEQ at a dose of 0.5 µg/kg/min at the EN initiation; therefore, the administration of EN was difficult. The H-NEQ group included patients with an intense inflammatory response, requiring a high NEQ dose (0.4 µg/kg/min) at EN initiation. The L-NEQ group patients had less severe disease, requiring the lowest exposure to NEQ doses (0.2 µg/kg/min) at EN initiation; most of the patients could tolerate EN. The conditions of M-NEQ group patients, requiring 0.4 µg/kg/min NEQ infusion, were intermediate between those of the H-NEQ and L-NEQ group patients.

The NEQ trajectories were correlated with the FI risk, which increased in the M-NEQ and H-NEQ groups, thereby requiring high NEQ doses. Moreover, most patients developed FI rapidly after EN initiation in the R-NEQ group. As compared to the M-NEQ group patients, the incidence of FI was insignificantly higher in the H-NEQ group patients for the same period. In our study, the incidence of suspected mucosal ischemia was 1.9% and confirmed mesenteric ischemia was 1% among all the patients, which was consistent with previous studies, including the NUTRIREA-2 study (2% in the enteral group) [26]. Moreover, the NUTRIVAD study reported a 4.5% incidence of suspected intestinal mesenteric ischemia and a 0.5% incidence of confirmed intestinal mesenteric ischemia in critically ill patients, requiring vasopressors [15]. In our study, there were two patients with suspected intestinal ischemia that were not confirmed, both of whom had severe respiratory and circulatory failure and died with rapid deterioration. Intestinal mesenteric ischemia is mainly associated with a low flow state, causing the non-occlusive mesenteric ischemia phenomenon. Studies reported that the use of inotropic drugs was a risk factor for mesenteric ischemia [12,15,35]. In the current study, the dose of dobutamine was the highest in the R-NEQ group, which had a significantly higher incidence of mesenteric ischemia (up to 11.76%) as compared to that in the other three groups. Therefore, in addition to vasopressors dosage, the use of inotropic drugs might also be an important concern in monitoring mesenteric ischemia in patients with shock.

The current study demonstrated that EN had more positive impacts on the patients in the L-NEQ, M-NEQ, and H-NEQ groups. Interestingly, except for those in the L-NEQ dose group, which showed a good EN tolerance, the patients in the other three groups showed poor EN tolerance. The L-NEQ group contained nearly half of the total patients included in this study. EN should be actively initiated if the patient can tolerate it. Clinicians are generally cautious about initiating EN in the H-NEQ and R-NEQ groups patients, especially in the latter group patients, who have a high risk of developing FI and serious gastrointestinal complications. Clinicians often consider delayed EN or trophic feeding for these patients; however, their proportion is very small as compared to all the patients. The M-NEQ group represented 40% of the overall patients. They lacked clinical identification and were less different from the L-NEQ group patients. Moreover, their clinical presentation and laboratory tests were not different from those of the L-NEQ group patients; the only difference was in their exposure to NEQ. This group patients are usually given aggressive treatment with EN. Our results suggested that these patients were not suitable for EN support. Therefore, the NEQ threshold should be selected as more appropriate for EN support in patients with shock when their NEQ is stable at 0.2 µg/kg/min and below.

This study described and analyzed the temporal trends in vasopressors in patients with shock for the first time. However, there were still certain limitations to this study. First, the sample size was small with an even smaller sample size in the H-NEQ and R-NEQ groups. Second, the patients were followed hourly with vasoactive drug data; however, all six-hour data were included for performing the latent category trajectory analysis, and the low-frequency follow-up might have missed the strong fluctuations in individual treatments. Therefore, the choice of a follow-up time window might have affected the interpretation of the results. Third, this was a post hoc analysis of a prospective study, and the results were exploratory and require further validation. Fourth, in this study, only the NEQ situation was followed up after EN initiation, and the NEQ dose before EN initiation was not recorded, which might have caused the loss of some important information. Fifth, the types of shock were not defined, and there might be heterogeneity. Sixth, the feeding rate and management of EN are also factors that affect the EN tolerance in patients with shock but were not included in the analysis. In the future, we need further high-quality multicenter cohorts of patient with shock to validate the results of this study.

## 5. Conclusions

This study identified four trajectories of NEQ in the patients with shock, which were characterized as L-NEQ, M-NEQ, H-NEQ, and R-NEQ doses. The FI risk was significantly associated with NEQ trajectories in the patients with shock. The results also suggested that the M-NEQ patients might be easily overlooked in clinical practices, and EN might be wrongly and aggressively initiated for these patients. Therefore, the therapeutic window of EN support below 0.2 µg/kg/min should be selected for NEQ stabilization. In addition, our findings may be more appropriate for the subgroup of patients with septic shock. Future high-quality studies are required to validate the effects of temporal changes in NEQ on EN tolerance and other clinical outcomes of patients with different types of shock.

## Figures and Tables

**Figure 1 nutrients-14-05393-f001:**
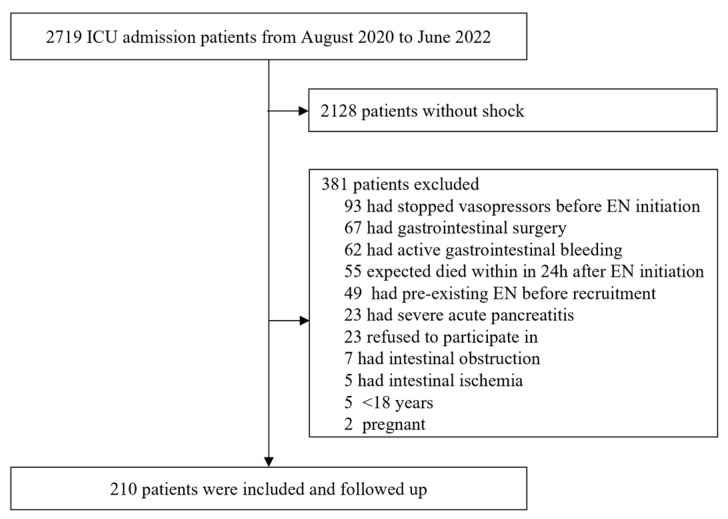
The enrollment and follow-up of patients. ICU, intensive care unit; EN, enteral nutrition.

**Figure 2 nutrients-14-05393-f002:**
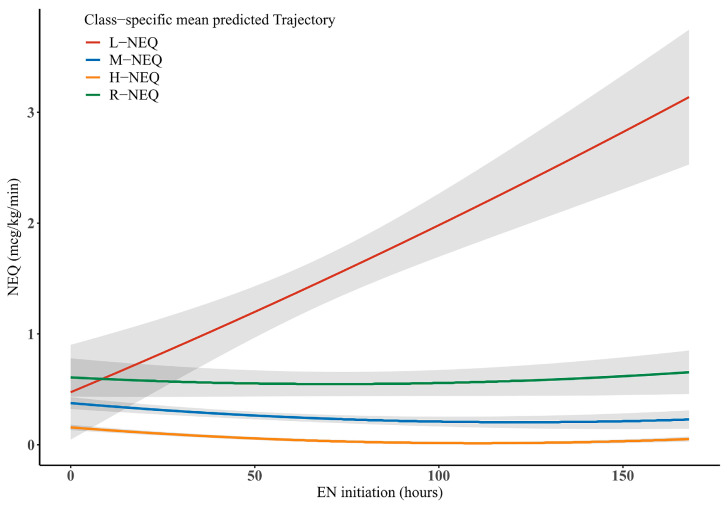
Trajectories of NEQ use in patients with shock after EN initiation. NEQ, norepinephrine equivalent dose; EN, enteral nutrition.

**Figure 3 nutrients-14-05393-f003:**
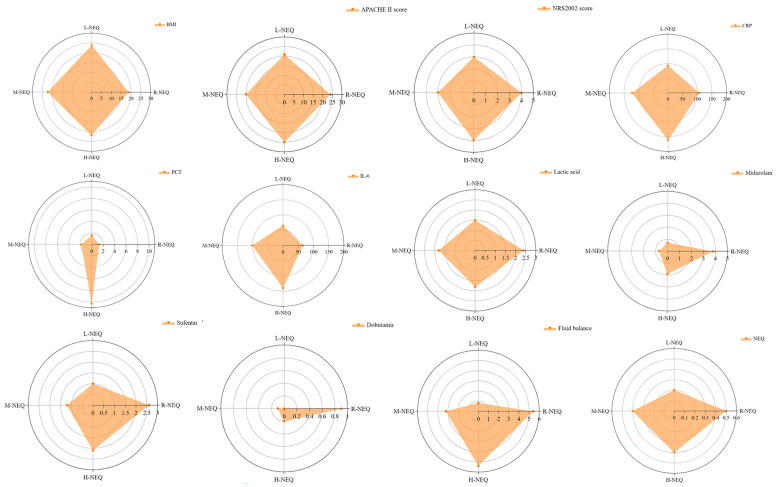
Radar charts of patient characteristics in the four groups. APACHE II, Acute Physiology and Chronic Health Evaluation II; BMI, body mass index; NRS2002, Nutritional Risk Screening 2002; CRP, C-reactive protein; NEQ, norepinephrine equivalent dose; PCT, procalcitonin; L-NEQ, low-dose stable NEQ; M-NEQ, moderate-dose stable NEQ; H-NEQ, high-dose stable NEQ; R-NEQ, rapidly rising NEQ.

**Figure 4 nutrients-14-05393-f004:**
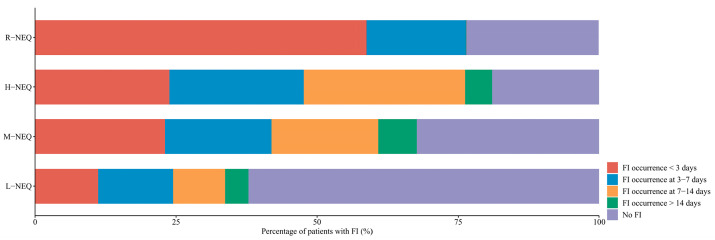
Stack graph of the percentage of patients with FI at different days in each group. FI, feeding intolerance; L-NEQ, low-dose stable NEQ; M-NEQ, moderate-dose stable NEQ; H-NEQ, high-dose stable NEQ; R-NEQ, rapidly rising NEQ.

**Figure 5 nutrients-14-05393-f005:**
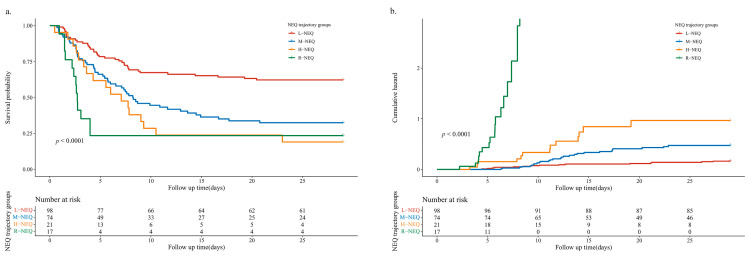
Kaplan–Meier survival curves (**a**) and cumulative risk (**b**) of FI for NEQ trajectory groups. NEQ, norepinephrine equivalent dose; L-NEQ, low-dose stable NEQ; M-NEQ, moderate-dose stable NEQ; H-NEQ, high-dose stable NEQ; R-NEQ, rapidly rising NEQ.

**Table 1 nutrients-14-05393-t001:** Baseline characteristics and follow-up by NEQ Trajectory Groups.

Characteristics	Total (*n* = 210)	L-NEQ (*n* = 98)	M-NEQ (*n* = 74)	H-NEQ (*n* = 21)	R-NEQ (*n* = 17)	*p* Value
Age (years), median (IQR)	64 (49.8–72)	63 (48–71.3)	64.5 (52–70)	67 (53.5–76.5)	62 (45.5–73)	0.462
Male Sex—No. (%)	141 (67.14)	66 (67.35)	47 (63.51)	14 (66.67)	14 (82.35)	0.526
BMI, median (IQR)	22.9 (20.2–24.8)	23.9 (21.5–25.8)	22.3 (19.9–24.7)	22 (19.5–23.8)	18.7 (17.2–23.5)	<0.001
APACHE II score, median (IQR)	20.9 (14–26)	20.9 (15–26)	20 (12.8–25.3)	25 (15.5–27.5)	24 (12–29)	0.548
NRS2002 score, median (IQR)	3 (3–4)	3 (3–4)	3 (3–4.3)	4 (3–5.5)	4 (3–6)	0.047
Diagnose—No. (%)						
Severe pneumonia	92 (43.81)	34 (34.69)	38 (51.35)	13 (61.9)	7 (41.18)	0.049
ARDS	20 (9.52)	10 (10.2)	4 (5.41)	3 (14.29)	3 (17.65)	0.339
MODS	29 (13.81)	14 (14.29)	8 (10.81)	2 (9.52)	5 (29.41)	0.223
Cardiac arrest	24 (11.43)	14 (14.29)	6 (8.11)	2 (9.52)	2 (11.76)	0.643
Cardiovascular disease ^1^	130 (61.9)	57 (58.16)	46 (62.16)	14 (66.67)	13 (76.47)	0.51
Neurological disease ^2^	85 (40.48)	42 (42.86)	30 (40.54)	10 (47.62)	3 (17.65)	0.226
Types of shock—No. (%)						
Sepsis shock	178 (84.76)	80 (81.63)	62 (83.78)	19 (90.48)	17 (100)	0.223
Cardiac shock	22 (10.48)	10 (10.2)	10 (13.51)	1 (4.76)	1 (5.88)	0.604
Hemorrhagic shock	25 (11.9)	13 (13.27)	10 (13.51)	2 (9.52)	0 (0)	0.429
Laboratory data, median (IQR)						
White blood cell (×10^9^/L)	10.6 (8–14.3)	10 (7.6–13.7)	11.5 (8.2–14.3)	13.7 (9.3–16.2)	10.4 (4.5–18.8)	0.124
Platelet (×10^9^/L)	97.5 (52–166)	95.5 (55.5–205.3)	109 (61–171)	87 (40–129.5)	72 (22.5–130.5)	0.122
Hemoglobin (g/L)	87 (76.8–100)	83.5 (72–98.3)	89 (78.8–100.3)	91 (76.5–103.5)	87 (74.5–118)	0.197
Albumin (g/L)	32.4 (29.4–35.8)	32.3 (29.3–36)	31.8 (28.6–34.8)	30.8 (28.8–37.3)	34.4 (32.7–37)	0.127
Total bilirubin (μmol/L)	17 (10.3–33.1)	16.9 (10.3–32.1)	17.9 (8.6–33.7)	15.9 (12.3–21.2)	27.6 (15.8–41.2)	0.223
Serum creatinine (μmol/L)	93 (60.8–152.3)	93 (62.8–164.5)	81.5 (57.8–150.8)	100 (72–142.5)	90 (63.5–124.5)	0.659
Glucose (mmol/L)	9 (6.6–12.2)	8.4 (6.5–11.4)	9.3 (6.7–13.5)	10.5 (8.7–12.6)	8.9 (7–11.8)	0.215
C-reactive protein (mg/L)	117.5 (65.2–167)	93 (56.4–136.3)	122.5 (69.1–172.3)	162 (123–236.5)	107 (39.3–178)	0.005
Procalcitonin (μg/L)	1.8 (0.6–9.4)	1.6 (0.4–6.9)	1.8 (0.7–7.2)	10.3 (0.8–21.7)	1.3 (0.7–8.2)	0.113
Interleukin-6 (μg/L)	74.5 (33.5–231.8)	64.8 (26.5–151.5)	101 (38.6–281.6)	140 (43.9–523.7)	64.9 (37–168.7)	0.102
Lactic acid (mmol/L)	1.7 (1.3–2.2)	1.5 (1.2–1.8)	1.8 (1.4–2.3)	1.8 (1.4–2.5)	2.4 (1.4–3.6)	<0.001
Arterial partial pressure of oxygen (mmHg)	98.9 (72.9–127.3)	104.4 (78.9–122.6)	98.3 (70.7–126.1)	93.5 (62.8–132.8)	77.9 (69.2–124.8)	0.569
Arterial oxygen saturation (%)	98.7 (95.6–99.4)	98.8 (96.5–99.6)	98.6 (96.2–99.3)	98.5 (91–99.3)	96.8 (93.8–99.3)	0.456
Sedation and analgesia, mean ± SD						
Midazolam (mg/kg/day)	1.1 ± 2.9	0.7 ± 1	0.7 ± 1.1	1.9 ± 2.6	3.8 ± 9.1	<0.001
Propofol (mg/kg/day)	5.3 ± 6.4	5.5 ± 6.7	4.3 ± 5.6	5.4 ± 5.5	8 ± 8	0.195
Dexmedetomidine (μg/kg/day)	2.3 ± 4.4	1.9 ± 2.1	3.1 ± 6.7	1.4 ± 2.4	2.9 ± 3.6	0.278
Remifentanil (μg/kg/day)	37.6 ± 60.8	33.8 ± 62.6	40.4 ± 56.8	44.1 ± 70.9	39.3 ± 58.2	0.856
Sufentanyl (μg/kg/day)	1.3 ± 2	1 ± 1.7	1.2 ± 1.9	2.1 ± 2.2	2.6 ± 2.9	0.004
Treatment—No.(%)						
Norepinephrine at EN initiation (μg/kg/min), median (IQR)	0.3 (0.1–0.5)	0.2 (0.1–0.3)	0.4 (0.2–0.6)	0.4 (0.3–0.6)	0.5 (0.4–0.7)	<0.001
NEQ at EN initiation (μg/kg/min), median (IQR)	0.3 (0.1–0.5)	0.2 (0.1–0.2)	0.4 (0.2–0.6)	0.4 (0.3–0.6)	0.5 (0.4–0.7)	<0.001
Prokinetics	94 (44.76)	53 (54.08)	28 (37.84)	10 (47.62)	3 (17.65)	0.019
CRRT	76 (36.36)	30 (30.93)	29 (39.19)	9 (42.86)	8 (47.06)	0.437
ECMO	13 (6.22)	9 (9.28)	2 (2.7)	0 (0)	2 (11.76)	0.144
Inotropic drugs use	85 (40.48)	22 (22.45)	36 (48.65)	16 (76.19)	11 (64.71)	<0.001
Dobutamine (daily dose) during the whole follow-up (μg/kg/day), mean ± SD	0.1 ± 0.9	0.0 ± 0.0	0.1 ± 0.9	0.2 ± 0.7	0.6 ± 2.3	0.091
Cumulative fluid balance in the first week (L), mean ± SD	2.5 ± 5.7	0.8 ± 4.3	3.2 ± 5.9	5.4 ± 8.9	5.4 ± 3.9	<0.001

All *p* values are two-tailed. Differences was analyzed among L-NEQ Group, M-NEQ Group, H-NEQ Group and R-NEQ Group. APACHE II, Acute Physiology and Chronic Health Evaluation II; ARDS, acute respiratory distress syndrome; BMI, body mass index; CRRT, continuous renal replacement therapy; ECMO, extracorporeal membrane oxygenation; IQR, interquartile range; MODS, multiple organ dysfunction syndrome; NRS2002, Nutritional Risk Screening 2002; NEQ, norepinephrine equivalent dose; SD, standard deviation; L-NEQ, low-dose stable NEQ; M-NEQ, moderate-dose stable NEQ; H-NEQ, high-dose stable NEQ; R-NEQ, rapidly rising NEQ. ^1^ Cardiovascular diseases are coronary heart disease, hypertension, acute myocardial infarction, infectious endocarditis, arrhythmia, cardiomyopathy, cardiac insufficiency, cardiac failure, and pericardial effusion. ^2^ Neurological diseases are Parkinson′s disease, hypoxic-ischemic encephalopathy, epilepsy, cerebral contusion, subarachnoid hemorrhage, intracranial infection, cerebral infarction, brain atrophy, intracranial hematoma, and stroke.

**Table 2 nutrients-14-05393-t002:** Clinical outcomes in patients with shock by NEQ Trajectory Groups.

Characteristics	Total (*n* = 210)	L-NEQ (*n* = 98)	M-NEQ (*n* = 74)	H-NEQ (*n* = 21)	R-NEQ (*n* = 17)	*p* Value
EN starting time from ICU admission (hours), median (IQR)	30.6 (18.1–54.2)	25.1 (17.8–46.4)	34.8 (17.1–60.1)	35.6 (18.7–72.3)	36.7 (20.2–71.8)	0.457
The EN intake at first week after EN initiation, median (IQR)						
Energy (kcal/kg)	8.6 (5.2–12.4)	8 (4.8–11.9)	9.9 (6.8–14.2)	7.7 (4.1–10.8)	5.5 (3.4–11)	0.017
Protein (g/kg)	0.3 (0.2–0.5)	0.3 (0.2–0.5)	0.4 (0.3–0.6)	0.3 (0.2–0.4)	0.2 (0.1–0.5)	0.014
Fat (g/kg)	0.2 (0.1–0.4)	0.2 (0.1–0.4)	0.3 (0.1–0.4)	0.2 (0–0.3)	0.2 (0–0.4)	0.465
The total nutrition intake at first week after EN initiation, median (IQR)						
Energy (kcal)	19 (13.6–24.7)	19 (13.9–24.8)	20.1 (14.2–25.1)	18.6 (10.9–23.9)	16.8 (10.6–23.1)	0.494
Protein (g)	0.6 (0.4–0.9)	0.7 (0.5–0.9)	0.6 (0.5–1)	0.6 (0.4–0.9)	0.3 (0.2–0.8)	0.043
Fat (g)	0.6 (0.4–0.9)	0.7 (0.4–0.9)	0.6 (0.4–0.9)	0.7 (0.3–0.9)	0.4 (0.2–0.9)	0.254
FI—no. (%)	117 (55.71)	37 (37.76)	50 (67.57)	17 (80.95)	13 (76.47)	<0.001
Diarrhea	34 (16.19)	9 (9.18)	17 (22.97)	6 (28.57)	2 (11.76)	
Vomiting/regurgitation	42 (20)	15 (15.31)	16 (21.62)	8 (38.1)	3 (17.65)	
Abdominal distension	46 (21.9)	17 (17.35)	17 (22.97)	4 (19.05)	8 (47.06)	
Ileus	8 (3.81)	3 (3.06)	4 (5.41)	1 (4.76)	0 (0)	
Suspected mesenteric ischemia	4 (1.9)	0 (0)	1 (1.35)	1 (4.76)	2 (11.76)	
Confirmed mesenteric ischemia	2 (1.0)	0 (0)	1 (1.35)	1 (4.76)	0 (0)	
Hospital infections—no. (%)	52 (24.88)	17 (17.53)	23 (31.08)	7 (33.33)	5 (29.41)	0.150
MV (day), median (IQR)	12 (6.5–21.1)	12.5 (5.5–23.1)	15 (9.9–25.3)	10.9 (5.4–17.6)	6.6 (5.1–7.7)	<0.001
ICU Length of stay (day), median (IQR)	15.8 (10.6–24)	17.5 (11.8–25.3)	17 (12–27.1)	12 (7–18.2)	7.6 (6–9)	<0.001
28-days mortality—no. (%)	73 (34.76)	15 (17.35)	28 (37.83)	13 (61.9)	17 (100)	<0.001

All *p* values are two-tailed. Differences was analyzed among L-NEQ Group, M-NEQ Group, H-NEQ Group and R-NEQ Group. EN enteral nutrition; FI feeding intolerance; NEQ, norepinephrine equivalent dose; ICU intensive care unit; IQR, interquartile range; MV mechanical ventilation; L-NEQ, low-dose stable NEQ; M-NEQ, moderate-dose stable NEQ; H-NEQ, high-dose stable NEQ; R-NEQ, rapidly rising NEQ.

**Table 3 nutrients-14-05393-t003:** Cox proportional hazard regression analysis of the effect of NEQ trajectory groups on FI.

FI	Model 1		Model 2		Model 3		Model 4	
	HR (95% CI)	*p* Value	HR (95% CI)	*p* Value	HR (95% CI)	*p* Value	HR (95% CI)	*p* Value
Baseline joint groups								
L-NEQ	1.0 (Ref)		1.0 (Ref)		1.0 (Ref)		1.0 (Ref)	
M-NEQ	2.226 (1.453, 3.409)	<0.001	1.895 (1.182, 3.038)	0.008	1.96 (1.214, 3.163)	0.006	1.963 (1.214, 3.177)	0.006
H-NEQ	2.974 (1.67, 5.296)	<0.001	2.384 (1.25, 4.545)	0.008	2.444 (1.279, 4.67)	0.007	2.317 (1.197, 4.484)	0.013
R-NEQ	4.258 (2.252, 8.053)	<0.001	3.374 (1.666, 6.833)	0.001	3.344 (1.653, 6.764)	0.001	3.146 (1.51, 6.554)	0.002
*p* for trend	<0.001		<0.001		<0.001		<0.001	
Covariates								
Age (years)			0.995 (0.983, 1.006)	0.369	0.992 (0.98, 1.005)	0.246	0.992 (0.979, 1.005)	0.238
BMI			0.983 (0.931, 1.036)	0.518	0.98 (0.929, 1.034)	0.463	0.984 (0.932, 1.039)	0.569
NEQ at EN initiation (μg/kg/min)			1.72 (0.794, 3.723)	0.169	1.632 (0.75, 3.554)	0.217	1.594 (0.729, 3.486)	0.243
APACHE II score					1.011 (0.987, 1.035)	0.389	1.01 (0.986, 1.035)	0.397
Dobutamine (μg/kg/day)							1.089 (0.926, 1.282)	0.302
Cumulative fluid balance at first week (L)							1.019 (0.986, 1.052)	0.258
Lactic acid (mmol/L)							0.944 (0.802, 1.111)	0.489

Model 1 was unadjusted. Model 2 was adjusted for age, BMI, and NEQ at EN initiation. Model 3 was adjusted for age, BMI, NEQ at EN initiation, and APACHE II score. Model 4 was adjusted for age, BMI, NEQ at EN initiation, APACHE II score, dobutamine, cumulative fluid balance at first week, and lactic acid at EN initiation. APACHE II: acute physiology and chronic health evaluation II, BMI: body mass index, CI: confidence interval, HR: hazard ratio, EN: enteral nutrition, FI: Feeding intolerance, NEQ: norepinephrine equivalent dose; L-NEQ, low-dose stable NEQ; M-NEQ, moderate-dose stable NEQ; H-NEQ, high-dose stable NEQ; R-NEQ, rapidly rising NEQ.

## Data Availability

The datasets used and/or analyzed during the current study are available from the corresponding author on reasonable request.

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
