# Peer review of "Association of Vasopressors Dose Trajectories with Enteral Nutrition Tolerance in Patients with Shock: A Prospective Observational Study"

_nutrients, 2022, doi:10.3390/nu14245393_

Round 1

Reviewer 1 Report

This is a very interesting study investigating the risk of feeding intolerance in critically ill patients with shock receiving vasopressors. Such studies are welcome since the issue in hand remains to be fully elucidated and clear guidelines are more or less lacking. Thus, the introduction and dosing of enteral nutrition in shock remains a conundrum. The present paper is well written and provides new additional data on this contradictory matter. I have no major comments.  

Author Response

Thanks very much for your suggestions. 

Reviewer 2 Report

I have read this paper with considerable interest, I find it well written and of a really interesting topic as well as very useful for clinical practice.  I only suggest inserting a figure that better explains the study protocol.

Author Response

Thanks very much for your rigorous consideration. We have added a figure on the simplified workflow of the study in Additional file 1: Figure S1 and described it in the section “2.5. Statistical analyses”.

Reviewer 3 Report

Luping Wang et al have performed a post-hoc analysis of a prospective observationnal study dealing with the link between vasopressor dose trajectories and feeding intolerance among critically ill patients with shock. Authors have identified 4 main vasopressors dose trajectories: low-dose stable, moderate-dose stable, high-dose stable, and rapidly rising dose. The incidence of feeding intolerance was 38%, 68%, 81% and 76% among this four groups, respectively. 

This is an interesting and well written study. The topic and the underlying question is of high importance: when could we safely introduce enteral nutrition among critically ill patients with shock ?

I have several remarks.

The characteristics of enteral nutrition products should be clarified (polymeric of semi-elemental; 1 kcal/ml or 1.5 kcal/ml...)

Regarding the "nutrition strategy", the first day, up to 150 ml/h was given to patients when feeding tolerance was considered good. This is very high, and higher than recommended by guidelines (full enteral nutrition should be reach progressively after 48-72 hours). Therefore, one could hypothesized that full EN was given the first day, and too rapidly. This might explain the very high rate of feeding intolerance in this study.  

Patients had mainly septic shock (85%). The conclusions identified in this study are possibly limited to the subgroup of patients with septic shock.

Figure 4 is very interesting. Feeding intolerance occuring the first week in the ICU accounted for 75% of the patients with rapidly rising dose, and approximately 50% of patients with moderate-stable or high-stable doses, and only 25% for "low dose" group.

Table 3. Regarding multivariate models, it could be interesting to include lactic acid, or inotropic drugs, with vasopressor dose trajectories. Indeed, need for inotropic drug could be an independent factor associated with feeding intolerance.

How many suspected acute mesenteric ischemia were finally confirmed ? Interestingly, no suspicion of acute mesenteric ischemia occured in the low dose group.

In the discussion section, acute mesenteric ischemia occurred in only 1% of the patients included in the NUTRIREA2 study, not 2%.

Author Response

The characteristics of enteral nutrition products should be clarified (polymeric of semi-elemental; 1 kcal/ml or 1.5 kcal/ml...)

Response:

Thanks very much for your suggestions. Both of polymeric (0.9 kcal/ml) and semi-elemental (0.8 kcal/ml) formula were used in our center. We have clarified the characteristics of enteral nutrition products in the manuscript in the section “2.3. Nutrition strategy”.

Regarding the "nutrition strategy", the first day, up to 150 ml/h was given to patients when feeding tolerance was considered good. This is very high, and higher than recommended by guidelines (full enteral nutrition should be reach progressively after 48-72 hours). Therefore, one could hypothesized that full EN was given the first day, and too rapidly. This might explain the very high rate of feeding intolerance in this study.

Response:

Thanks very much for your rigorous consideration. The feeding rate in the original article was misrepresented; in our clinical practice, the feeding rate did not exceed 150 ml/h. We revised the statement to "but did not exceed 150 ml/h" in the section “2.3. Nutrition strategy”.  The feeding rate and management of enteral nutrition are also the most important factors affecting feeding tolerance in critically ill patients; however, the feeding rate was not collected in our current study, which might have caused the loss of some important information. We added and discussed this limitation in the section “Discussion”. In our follow-up work, we will consider collecting information related to feeding rate and other relevant factors for related studies.

Patients had mainly septic shock (85%). The conclusions identified in this study are possibly limited to the subgroup of patients with septic shock.

Response:

We have described in the section “Conclusion” that the conclusions identified in this study might be more appropriate to patients with septic shock.

Table 3. Regarding multivariate models, it could be interesting to include lactic acid, or inotropic drugs, with vasopressor dose trajectories. Indeed, need for inotropic drug could be an independent factor associated with feeding intolerance.

Response:

We have included lactic acid and dobutamine (μg/kg/d) (the main inotropic drug used in our cohort) into the Cox proportional hazard regression analysis of the effect of NEQ trajectory groups on FI and 28-days mortality (both were Model 4), the results were shown in the Table 3 and Table S4.

How many suspected acute mesenteric ischemia were finally confirmed? Interestingly, no suspicion of acute mesenteric ischemia occurred in the low dose group.

Response:

Only two patients were finally confirmed acute mesenteric ischemia through abdominal computerized tomography. The other two patients developed severe respiratory and circulatory failure and died of rapidly deteriorating conditions, so no confirmation was made. We added the criterion of confirmed mesenteric ischemia in the section “2.4. Outcomes measurement”, described the information of confirmed acute mesenteric ischemia in the Table 2 and discussed it in the section “Discussion”.

In the discussion section, acute mesenteric ischemia occurred in only 1% of the patients included in the NUTRIREA2 study, not 2%.

Response:

Thank you very much for your rigorous consideration. We have qualified this information in the section "Discussion". In the NUTRIREA2 study, acute mesenteric ischemia occurred in only 1% of the whole cohort, and 2% in the enteral group.